[Supplementary Material · pytorch_paper_appendix.pdf]

# Appendix: benchmarking methodology

## A   Hardware

We ran all our benchmarks with the following hardware configuration:

- 2 Intel Xeon E5-2698 v4 @ 2.20GHz (20 cores per socket)
- 1 NVidia Quadro GP100
- Ubuntu 18.0.4

## B   Software environment

To run our benchmarks we used the latest stable versions of MxNet, PyTorch, and TensorFlow, as well as the latest revision of the NVidia toolchain. At the time we started our experiments (late April 2019), this corresponded to the following configuration:

**Table 1:** ToolChain.

| Python | Cuda | Cudnn | MxNet | PyTorch | Tensorflow |
|--------|----------|-------|-------|---------|------------|
| 3.6.8  | 10.0.130 | 7.3.1 | 1.4.0 | 1.0.1   | 1.13.1     |

To level the playing field, we installed the three frameworks in a single unified benchmarking environment: this ensured that they had access to a the exact same set of tools and libraries. We didn't want to risk skewing the results by allowing a framework to install a different version of the C library for example.

We leveraged anaconda3 and pip to create this benchmarking environment. The exact commands and packages we used are reproduced in listing 1.

**Listing 1:** Environment setup.

```
# Install the ml frameworks
conda install tensorflow-gpu pytorch
pip install mxnet-cu100mkl
# Install misc tools the benchmarks depend on
conda install torchvision opencv scikit-image
pip install tqdm gluonnlp fairseq nltk==3.2.5
```

## C   Benchmark setup

Every tools relies on its own language to describe and train neural network architectures, and depends on manual optimizations to be able train models efficiently. Therefore to be as fair as possible when comparing the performance of various frameworks, we had to be sure that we were training the same model architecture (e.g. same number of layers, same number of parameters, and so on...) while leveraging setups optimized for every framework. In order to do so, we chose 6 common models, and downloaded the best implementation of these models we could find for each framework. In particular, we used the implementations shipped with the frameworks themselves whenever possible.

We ended up with the following models: AlexNet, VGG-19, ResNet-50, MobileNetV2, GnmtV2, and NCF. We provided a detailed listing of the steps we took to run these models in listings 2, 3, 4, 5, 7, and 6.

## D   Data collection and analysis

We ran the scripts listed in 2, 3, 4, 5, 7, and 6 4 times on 3 distinct machines with the same hardware and software configuration. We parsed the log files to extract the throughput numbers reported by the

tools, being careful to discard any number reported during the first 2 minutes of training, since they are often skewed due to one time setup, library initialization, and so on. We then derived the mean and the 95% confidence interval from this data.

**Listing 2:** Setup for the AlexNet benchmarks.

```
# For MxNet
git clone https://github.com/apache/incubator-mxnet/
cd incubator-mxnet/example/image-classification
python train_imagenet.py --gpus="0" --benchmark=1 --network=alexnet
# For PyTorch
cd /tmp
git clone https://github.com/pytorch/vision
cd vision/references/classification
export PYTHONPATH=$PYTHONPATH:/tmp/vision
python train.py --model=alexnet -b 128
# For TensorFlow
https://github.com/tensorflow/models
cd models/tutorials/image/alexnet
python alexnet_benchmark.py
```

**Listing 3:** Setup for the VGG-19 benchmarks.

```
# For MxNet
git clone https://github.com/apache/incubator-mxnet/
cd incubator-mxnet/example/image-classification
python train_imagenet.py --gpus="0" --benchmark=1 --network=vgg --num-layers=19
# For PyTorch
cd /tmp
git clone https://github.com/pytorch/vision
cd vision/references/classification
export PYTHONPATH=$PYTHONPATH:/tmp/vision
python train.py --model=vgg19 #-b 128
# For TensorFlow
cd /tmp
https://github.com/tensorflow/models
cd models/research/slim/
bazel build download_and_convert_imagenet
bazel-bin/download_and_convert_imagenet /tmp/imagenet
python train_image_classifier.py  --train_dir=/tmp/train  --dataset_name=imagenet \
    --dataset_split_name=train --dataset_dir=/tmp/imagenet --model_name=vgg_19
```

**Listing 4:** Setup for the ResNet-50 benchmarks.

```
# For MxNet
git clone https://github.com/apache/incubator-mxnet/
cd incubator-mxnet/example/image-classification
python train_imagenet.py --gpus="0" --benchmark=1
# For PyTorch
cd /tmp
git clone https://github.com/pytorch/vision
cd vision/references/classification
export PYTHONPATH=$PYTHONPATH:/tmp/vision
python train.py --model=resnet50
# For TensorFlow
cd /tmp
https://github.com/tensorflow/models
cd models/official/resnet
export PYTHONPATH=$PYTHONPATH:/tmp/models
python imagenet_main.py --use_synthetic_data
```

**Listing 5:** Setup for the MobileNetV2 benchmarks.

```
# For MxNet
git clone https://github.com/apache/incubator-mxnet/
cd incubator-mxnet/example/image-classification
python train_imagenet.py --gpus="0" --benchmark=1 --network=mobilenetv2 \
    --batch-size=32
# For PyTorch
cd /tmp
git clone https://github.com/pytorch/vision
cd vision/references/classification
export PYTHONPATH=$PYTHONPATH:/tmp/vision
python train.py --model=mobilenet_v2 # 3-b 128
# For TensorFlow
cd /tmp
export DATASET_DIR=/tmp/imagenet
export TRAIN_DIR=/tmp/inception
git clone https://github.com/tensorflow/models
cd models/research/inception/slim
bazel run download_and_convert_imagenet $DATASET_DIR
bazel run -c opt train_image_classifier --  --train_dir=${TRAIN_DIR} \
    --dataset_name=imagenet --dataset_split_name=train \
    --dataset_dir=${DATASET_DIR} --model_name=mobilenet_v2
```

**Listing 6:** Setup for the NCF benchmarks.

```
# For PyTorch
git clone https://github.com/NVIDIA/DeepLearningExamples
cd DeepLearningExamples/PyTorch/Recommendation/NCF
docker build . -t nvidia_ncf
bash scripts/docker/interactive.sh
mkdir data
docker run --runtime=nvidia -it --rm --ipc=host \
    -v ${PWD}/data:/data nvidia_ncf bash
./prepare_dataset.sh
python -m torch.distributed.launch --nproc_per_node=1 ncf.py \
    --data /data/cache/ml-20m
# For TensorFlow
git clone https://github.com/NVIDIA/DeepLearningExamples
cd DeepLearningExamples/TensorFlow/Recommendation/NCF
docker build . -t nvidia_ncf
mkdir data
docker run --runtime=nvidia -it --rm --ipc=host \
    -v ${PWD}/data:/data nvidia_ncf bash
./prepare_dataset.sh
numgpu=1
datadir=/data/cache/ml-20m
mpirun -np $numgpu --allow-run-as-root python ncf.py --data $datadir
```

**Listing 7:** Setup for the GNMTv2 benchmarks.

```
# For PyTorch
git clone https://github.com/NVIDIA/DeepLearningExamples
cd DeepLearningExamples/PyTorch/Translation/GNMT
bash scripts/docker/build.sh
bash scripts/docker/interactive.sh
bash scripts/wmt16_en_de.sh
python3 -m launch train.py --seed 2 --train-global-batch-size 1024
# For TensorFlow
git clone https://github.com/NVIDIA/DeepLearningExamples
cd DeepLearningExamples/TensorFlow/Translation/GNMT
bash scripts/docker/build.sh
bash scripts/docker/interactive.sh
bash scripts/wmt16_en_de.sh
python nmt.py --output_dir=results --batch_size=192 --learning_rate=8e-4
```