[Reviews · NeurIPS 2019]

Reviewer 1



*Rebuttal* I thank the authors for addressing my comments and including the additional performance comparison. I will increase my score to 9. *Summary* PyTorch is an open-source deeplearning library that strives to marry good performance, flexibility and usability. It is specifically designed for researchers with the goal to enable easy experimenting with new features. Through seamless integration in the Python ecosystem it enables the interoperability with other python libraries which makes prototyping easy for the user. One may argue whether a systems/software paper, presenting the implementation details of a library should be published at NeurIPS, or whether it would be a better fit for USENIX or SysML. However, given the impact of the library in the research community I strongly support the publication of this paper at NeurIPS. PyTorch is specifically designed for the research community, thus of high interest to most attendees. The library offers what a researcher needs to validate new ideas without large implementation effort. By being fully open-source and providing flexibility to the user, it enables the exploration of new features, including new models and performance optimizations. Thus, it has high value to the community and helps drive research forward. [clarity] The paper is well written and easy to follow. It touches upon many challenges faced when implementing such a framework and gives some insights into its power and also its limitations. *Comments* I feel from the paper it is not clear whether PyTorch can run in the absence of GPUs. It says that it supports GPUs, but does it depend on the presence of GPUs? My main criticism is that you claim that PyTorch offers performance comparable to the fastest current libraries for DL. But what are these libraries? In Section 6 you compare to MXNet and Tensorflow. I feel to support this claim a broader comparison to other libraries would be necessary, or at least some evidence/reference that Tensorflow and MXNet are currently the fastest out there. *Minor comments* - I think it is nice to have a dedicated publication for PyTorch so it is more clear in the future how to cite it. However I wonder how attribution of this open-source library is handled. - The authors could try to tight the paper a bit more to the conference. There is only a single reference that has been published at NeurIPS and maybe there could be done a better link. - Line 87: remove ‘.’

Reviewer 2



This paper presents the PyTorch deep learning library, with a major focus on design principles and design choices made in the creation of the software. From what I can tell, the paper presents the first peer-reviewed submission of the full PyTorch software library (a previous submission to a NIPS autodiff workshop, https://openreview.net/forum?id=BJJsrmfCZ, seems to detail only the automatic differentiation component). I applaud the authors for focusing their submission around their key design principles and decisions, rather than presenting a detailed tutorial of the software's use. This distinguishes the submission from a "white paper," i.e., a paper which reports on how to use a piece of software while marketing its merits, but having less scientific value. Instead, the paper reads as a kind of manifesto for what deep learning software should aspire to be, while also detailing how this vision can be achieved through careful design and implementation. Design decisions are carefully mapped out, and justifications are given for why a certain choice was made, always referencing back to the basic design principles (in particular, usability and performance). The paper is well thought-out, very clear, and carefully written. There is no one big idea here. Ideas emerging from previous works are clearly stated. Nor is there any new or surprising result here, since PyTorch has been available (and widely used) for several years in the community. However, the paper demonstrates the successful synthesis of several key ideas in the field of deep learning software. There are a lot of smaller ideas in the implementation which, while not flashy, fit together to give the library overall strong performance. The authors detail the consequences and tradeoffs of their design choices carefully, making clear that sometimes this may lead to slightly worse performance, which is acceptable if it makes the software easier to use. There was no guarantee that PyTorch would be competitive with other deep learning libraries based on static dataflow graphs. It could well have been that PyTorch forced users to make a big tradeoff between performance and ease of programmability. Due to the careful design and implementation of the library, it appears that PyTorch is on par performance-wise with other graph-based libraries, despite efficiency being the not-quite-primary design principle. The developers should be acknowledged for taking this risk. I also appreciate that, while the authors focus the paper on outlining their design principles and decisions, they also provided some evaluations which, with limited space, largely confirm the validity of their (sometimes technical) design choices. These are presented in a variety of ways: benchmarks, comparisons, code profiling, and gauging user adoption. The evaluations were very clear. For instance, I have no experience in GPU memory management, so it took some concentration to follow along Sec. 5.3 carefully; yet the accompanying evaluation in Sec. 6.2 made the consequences of PyTorch's custom memory allocator very clear. Overall, PyTorch is a significant contribution to the field of deep learning software, influencing not just the researchers who use the library, but also designers of other scientific software (and likely to have lasting influence beyond this). As such, this paper deserves recognition as one of the top papers at NeurIPS.

Reviewer 3



Since Pytorch has been used and tested by the community for years, the validity of the paper is without a doubt. The reviewer finds the paper easy to read and learns a few new things about PyTorch. It is very certain that this paper has high originality, quality, clarity, and significance, though it might come a bit late. In lines 37 - 38, maybe it is better to reorder Python, C++, Lisp, and Lua, such that they match with Numpy, Torch, Eigen, and Lush. ===== After authors responses ====== Thanks for addressing the question.

[Author Response · NeurIPS 2019]

We wish to thank our reviewers for their insightful feedback that helped us improve the clarity and overall quality of our work. We have revised the paper as suggested by the reviewers. The corresponding changes are:

- As mentioned by reviewer #1, the hardware requirements of PyTorch were not clearly explained. We therefore updated section 5.1 of our paper to state that PyTorch can run in a CPU only environment but will happily take advantage of GPUs if such devices are available.

- To better support our performance claims, we compared the speed of PyTorch against that of 3 additional libraries (namely Chainer, CNTK and PaddlePaddle) as requested by reviewer #1. We included in our new comparison matrix the tools that Amazon or NVidia support officially. In essence, we relied on these companies to perform a market research study on our behalf and identify the best solutions currently available. However, we excluded Caffe and Theano from the final list since they are no longer actively maintained. We also excluded Keras since it is essentially a frontend running on top of TensorFlow, CNTK, or Theano. We reproduced the updated version of Table 1 below. We believe that these additional results confirm that the performance of PyTorch is indeed very competitive.

- Reviewer #1 wondered about attribution. Since more than 1000 people have contributed to PyTorch, we can't list every single individual. Instead we've highlighted the people who had a profound impact on the library and have significantly helped shape its development. We've also acknowledged the impact of the community as a whole in section 8.

- We fixed the typographical error line 87 pointed out by reviewer #1.

- We realized that didn't cite the "Automatic differentiation in PyTorch" work from Paszke, Gross, Chintala, Chanan, Yang, DeVito, Lin, Desmaison, Antiga and Lerer submitted at the 2017 NIPS autodiff workshop. We added the missing reference in section 4.3.

- Reviewer #2 mentioned that the last paragraph of Section 5.2 is confusing. We agree, and rewrote the paragraph to emphasize that we keep the execution of tensor operators on CPU synchronous since, unlike on GPU, the benefits of an asynchronous execution model are overshadowed by the overhead of cross-thread communication and synchronization.

- We rewrote section 5.4 to explicitly state that the PyTorch multiprocessing module builds upon the native Python multiprocessing library. We emphasized that while the PyTorch version of the module optimizes the transfer of large tensors between processes, it leverages the Python implementation whenever possible.

- We updated Listing 1 to refer to the 'nn.functional' module instead of 'F' to increase the clarity of the code as suggested by reviewer #2

- We reordered the list of programming languages to match the order in which array libraries are listed line 37-38 to improve clarity as suggested by reviewer #3.

| Framework | Throughput (higher is better) | | | | | |
| | AlexNet | VGG-19 | ResNet-50 | MobileNet | GNMTv2 | NCF |
| --- | --- | --- | --- | --- | --- | --- |
| Chainer | $778 \pm 15$ | N/A | $\mathbf{219} \pm 1$ | N/A | N/A | N/A |
| CNTK | $845 \pm 8$ | $84 \pm 3$ | $210 \pm 1$ | N/A | N/A | N/A |
| MXNet | $\mathbf{1554} \pm 22$ | $113 \pm 1$ | $218 \pm 2$ | $444 \pm 2$ | N/A | N/A |
| PaddlePaddle | $933 \pm 123$ | $112 \pm 2$ | $192 \pm 4$ | $\mathbf{557} \pm 24$ | N/A | N/A |
| TensorFlow | $1422 \pm 27$ | $66 \pm 2$ | $200 \pm 1$ | $216 \pm 15$ | $9631 \pm 1.3\%$ | $4.8e6 \pm 2.9\%$ |
| PyTorch | $1547 \pm 316$ | $\mathbf{119} \pm 1$ | $212 \pm 2$ | $463 \pm 17$ | $\mathbf{15512} \pm 4.8\%$ | $\mathbf{5.4e6} \pm 3.4\%$ |

**Table 1:** Training speed for 6 models using 32bit floats. Throughput is measured in images per second for the AlexNet, VGG-19, ResNet-50, and MobileNet models, in tokens per second for the GNMTv2 model, and in samples per second for the NCF model. The fastest speed for each model is shown in bold.

[Meta-Review · NeurIPS 2019]

PyTorch is a useful and impactful package, and has made some interesting and influential design decisions. This paper explains the philosophy behind some of these decisions. Although the framework is not new this year, we still expect broad interest in this exposition, and that having an archival paper will be useful for the community to cite. Given the subject of this paper, the style is different from a typical NeurIPS submission and it needed to be evaluated differently. In particular, many of the claims in the paper are not convincingly tested or demonstrated in the paper as we would normally require. Instead, the utility of the ideas has been demonstrated through wide-spread adoption and impact in the NeurIPS community and beyond. Like the JMLR open source software track guidelines, existing (rather than potential) community engagement was important for an acceptance decision here. Newer software projects may need to use more of the body of their papers to test their claims. In addition to the reviews, the following critical comments were gathered from an area chair during the review process, which may help improve the paper, or be useful to consider going forwards: * Claims about ease of implementation could be tested with a user study. Alternatively a paper could provide examples of algorithms that are difficult to implement in competing packages. (These are also ways that a less-established framework could convince reviewers.) * The goal of "being Pythonic", is broad and subjective, and so not everyone will agree it has been met. Implementing each function as a "module" class, and having two different ways to do this, can be viewed as un-Pythonic by some users. * The figures are low-resolution bitmaps and would appear more professional if included in a vector format, or at least as crisper bitmaps. * Given the central role of derivative computations, there could be a deeper treatment: - There's no mention of the more general vector-Jacobian products that reverse-mode autodiff allows, or the computational complexity of different alternatives. - There's no discussion of nested autodiff, which is a difficult-to-get-right feature. * There are naming choices in PyTorch that are questionable: - Calling arrays "tensors" (not unique to PyTorch). - "Autograd" isn't (or wasn't) an accepted generic term for autodiff, but the name of a competing package.